# Morphological Characteristics of Genital Organ-Associated Lymphoid Tissue in the Vaginal Vestibule of Goats and Pigs

**DOI:** 10.3390/vetsci10010051

**Published:** 2023-01-11

**Authors:** Tsolmon Chuluunbaatar, Osamu Ichii, Md. Abdul Masum, Takashi Namba, Yasuhiro Kon

**Affiliations:** 1Laboratory of Anatomy, Department of Basic Veterinary Sciences, Faculty of Veterinary Medicine, Hokkaido University, Hokkaido 060-0818, Japan; 2Department of Basic Science of Veterinary Medicine, School of Veterinary Medicine, Mongolian University of Life Science, Ulaanbaatar 17024, Mongolia; 3Laboratory of Agrobiomedical Science, Faculty of Agriculture, Hokkaido University, Hokkaido 060-8589, Japan; 4Department of Anatomy, Histology, and Physiology, Faculty of Animal Science and Veterinary Medicine, Sher-e-Bangla Agricultural University, Dhaka 1207, Bangladesh

**Keywords:** genital organ-associated lymphoid tissue, mucosa-associated lymphoid tissue, goat, pig, vaginal vestibule, lymphatic nodule, diffuse lymphoid tissue

## Abstract

**Simple Summary:**

Mucosa-associated lymphoid tissue is a diffuse system of aggregations of lymphoid tissues found in the gut, nasopharynx, bronchus, tonsils, eye, skin, salivary glands, and urinary tract, which are in close contact with the outer environment. We previously reported the presence of lymphoid tissue in the mucosa of the vaginal vestibule in cows, but information regarding other animals is scarce. This study investigated the gross anatomical and histological features of lymphoid tissue in the mucosa of the vaginal vestibule of goats and pigs. The vaginal vestibules of healthy adult goats and pigs were examined using hematoxylin and eosin staining, immunohistochemistry, and immunofluorescence. Furthermore, scanning electron microscopy was used for analyzing the ultrastructure of the mucosal surface. Goat and pig lymphoid tissues comprised B cells, T cells, macrophages, dendritic cells, and plasma cells. Furthermore, the vaginal vestibule epithelium covering lymphoid tissues was disrupted by a large intercellular space in both goats and pigs, and immune cells directly contacted the luminal space through this area lacking epithelium. No obvious differences were found between goats and pigs, suggesting that genital organ-associated lymphoid tissue is a conserved local immunological barrier in these animals.

**Abstract:**

Mucosa-associated lymphoid tissue (MALT) is a specialized form of peripheral lymphoid tissue (LT), which is found on mucosal surfaces exposed to the environment. However, morphological data of these tissues in farm animals are scarce. This study investigated the gross anatomical and histological features of genital organ-associated lymphoid tissues (GOALTs) in the vaginal vestibule (VV) of healthy, non-pregnant, adult goats and pigs. Their VVs were composed of stratified squamous, non-keratinized epithelium, and various-sized dark-blue hematoxylin-positive spots were observed in whole-mount specimens, which were diffusely distributed throughout the mucosal surfaces. These spots were histologically identified as LTs and consisted of lymphatic nodules (LNs) or diffuse lymphoid tissue (DLTs). Both LNs and DLTs contained B cells, T cells, macrophages, dendritic cells, plasma cells, and high endothelial venules. Only the numbers of B cells were significantly higher in both the LNs and DLTs of pigs compared to goats. Furthermore, the surface of the VV epithelium covering the LTs was partially disrupted with a large intercellular space containing abundant connective tissue fibers with numerous lymphocytes. In conclusion, GOALTs in the VV appear to be common local immunological barriers in both examined animals. This knowledge is crucial for understanding the structures and disorders of female reproductive organs in farm animals.

## 1. Introduction

Mucosa-associated lymphoid tissue (MALT) is broadly distributed on mucosal surfaces and found at strategic sites for efficient antigen sampling [1]. MALTs are located close to the mucosal epithelium and consist of structured lymphoid tissue (LT), which can comprise single or several aggregated LNs. Recirculating lymphocytes enter the MALT through high endothelial venules (HEV) in the interfollicular area. Furthermore, lymphocytes and dendritic cells infiltrate the epithelium overlying the MALT [1]. In aquatic and terrestrial animals, MALTs are classified according to their anatomical location as follows: lymphoid tissues of Waldeyer’s ring—lingual, palatine, paraepiglottic, pharyngeal, and tubal tonsils and the tonsil of the soft palate [2]; conjunctiva-associated lymphoid tissue, nose or nasopharynx-associated lymphoid tissue; bronchus-associated lymphoid tissue, gut-associated lymphoid tissue (GALT), larynx- and trachea- associated lymphoid tissue, (salivary) duct-associated lymphoid tissue, lacrimal duct-associated lymphoid tissue, and genital organ-associated lymphoid tissue (GOALT) [2,3,4,5,6].

MALTs in pigs have been studied by several researchers because of their similarities to humans. The porcine large intestine harbors GALT, including multi-follicular submucosal lymphoid clusters, as well as smaller mucosal isolated LNs that contain high numbers of T and B cells [7]. BALTs are mostly found at bronchiolar bifurcations as a single dome-shaped LN, which bulges out from the mucosal surface into the airway [8]. The size of NALT found on the roof of the nasopharynx in pigs varies from 2 to 4 cm. Similarly to pigs, GALT, BALT, and NALT have also been identified in goats. GALTs are densely found in the distal colon and rectum as solitary LNs with round tubercles and a central depression. In goats, BALT differs from other locally organized tissues, is not developed during the prenatal period, and is fully affected by antigenic stimuli [9].

The vaginal vestibule (VV) is the terminal part of the genital tract, and the urethra opens the border between the vagina and the VV. In adult pigs, the VV is relatively long since the urethra enters the genital tract rather far cranially, and the external urethral orifice is associated with a small suburethral diverticulum [10]. In adult goats, the VV slopes ventrally to the opening between the labia of the vulva and is less distensible than the vagina [10]. The VV of the goat is 2.5–3.0 cm in length, and the major vestibular glands are not permanent structures; whereas, the VV of the pig is comparatively longer (approximately 7.5 cm) and does not contain any major vestibular glands [11,12]. The mucosal epithelium of VV is composed of non-keratinized stratified squamous epithelium and forms longitudinal folds called rugae in both goats and pigs [13].

Both innate and adaptive factors are involved in the immune system of reproductive organs. The innate system in the female genital tract is comprised of epithelial cells, which provide a strong physical barrier through their tight junctional function, and secrete antibacterial compounds, cytokines, and chemokines for recruiting and activating immune cells [14]. The adaptive immune system encompasses pathogen-specific defense mechanisms, such as antigen-presenting cells that mediate the cellular immune response by processing and presenting antigens for recognition by certain lymphocytes. Importantly, both innate and adaptive humoral mediators and immune cells in the female genital organ are regulated by sex hormones and, therefore, fluctuate through estrous cycles [15,16,17].

The genital organs of women and the female individuals of various species contain MALT during the normal or pathological changes. Several studies have been conducted on GOALTs to assess the effects of medicine delivery and mucosal vaccines on the female genital tract of human or non-human primates, cows, and pigs [18,19,20]. Interestingly, Holstein Friesian cows showed abundant LTs in their mucosae of the VV and clitoris, which consist of LN and diffuse lymphoid tissue (DLT) [21]. Additionally, the Guiana dolphin (*Sotalia guianensis*) is a representative aquatic vertebrate that exhibits LT aggregation in the lamina propria (LP) of the uterus [22]. Furthermore, the wild-type female mouse does not contain MALT in the external part of the genital organ (EPGO), but MALT-like structures have been found in the mucosa of the vagina after intravaginal inoculation with particular peptides [23]. Regarding the human genital organ, LTs are located in the mucosa of the uterus of healthy women and locally organized LTs are observed in the VV of women with provoked vulvodynia [24,25].

A clear understanding of the species-specific features of GOALT is important for comparative research in order to understand the border between normal and pathological conditions in EPGO and the development of therapeutic methods such as mucosal vaccination. This study aimed to describe the morphology of GOALT in the VV of goats and pigs during adulthood.

## 2. Materials and Methods

### 2.1. Sample Preparations from Healthy Goats and Pigs

All animal handling protocols and procedures were carried out according to the Institutional Animal Care and Use Committee of the Faculty of Veterinary Medicine, Hokkaido University (approval No.: 19-0097). The experiments on animals were conducted following the guidelines for the Care and Use of Laboratory Animals, Graduate School of Veterinary Medicine, Hokkaido University, Faculty of Veterinary Medicine. Hokkaido University (approved by the Association for Assessment and Accreditation of Laboratory Animal Care International). Over one-year-old healthy and non-pregnant Saanen breed (*Capra hircus*) goats (total *n* = 8) were sedated with xylazine (3.0 mg/kg body weight) through an intramuscular or intravascular route. Over one-year-old mixed breed pigs (total *n* = 8) were anesthetized with intramuscular injections of a combination of medetomidine hydrochloride (0.04 mg/kg body weight), midazolam (0.2 mg/kg body weight), and ketamine hydrochloride (20 mg/kg body weight). All the animals were deeply anesthetized with injections of pentobarbital sodium (500 mg/kg body weight) administered via the intravascular route. The method of sacrifice of the animals, tissue fixation, and tissue sampling were carried out as previously described [21].

### 2.2. Vaginal Smear Observation in Goats

To determine the estrus stages, vaginal smears were collected from each goat at 1-day intervals using disposable sterile cotton vaginal swabs wetted with 0.01 M phosphate-buffered saline (PBS). The vulva and perineum were gently wiped with wet cotton pads prior to each examination. The swab was gently inserted into the anterior vagina to a depth of 5 cm. After rolling the swab a few times in the vagina, it was immediately inserted into tubes containing 15 mL 0.9% sodium chloride and centrifuged for 20 min at 1885× *g* (Kubota 8800; Kubota, Tokyo, Japan). The separated upper fraction of the liquid was discarded, and 500 μL of PBS was added for dilution. Ten microliters of the diluted solution were inserted into a hemocytometer counting chamber. The cells encountered in the vaginal smear were categorized as parabasal, intermediate, and superficial cells, and their proportions were determined. Vaginal epithelial cells were identified using an inverted phase-contrast microscope (CK30; Olympus Optical, Tokyo, Japan), based on their morphological characteristics. The percentage of vaginal cells was calculated as the number of each cell type divided by the total number of cells counted in four microscopic fields. The same amount of diluted cell suspension was smeared on a glass slide, dried, and stained with Diff-Quik (Sysmex, Hyogo, Japan). Briefly, specimens were dipped in fixative solution (light blue) around ten seconds, then in solution I (red) and solution II (dark blue) for five seconds, respectively. Next, specimens were slightly washed with distilled water, dried, and mounted with cover glass.

### 2.3. Determination of Vaginal pH and Temperature in Goats

For goats, the vaginal pH level was measured using pH paper (Advantec, Tokyo, Japan) with a 5.8–8.2 indicator. The pH paper was gently inserted into the vagina, 5 cm deep, and soaked in vaginal mucus. The color change of the pH paper was compared with the attached standard color. Vaginal temperature was measured using a thermometer MC-672L (Omron Healthcare, Kyoto, Japan) with an accuracy of ±0.05 °C. A thermometer was inserted 5 cm deep into the vagina, and the results were recorded.

### 2.4. Whole-Mount Observation

The collected EPGOs of each animal were flattened with pins, with the mucosa on the upper side, soaked in Mayer’s hematoxylin for 8–10 min to visualize LTs, and examined at 3–4 min intervals, as described previously [21].

### 2.5. Histological Analysis

The VV of each animal was divided into more than five sections based on their size. The specimens were routinely dehydrated with ethanol and then embedded in paraffin. Paraffin sections (4–5 μm) were stained with hematoxylin and eosin (HE). The stained sections were analyzed using a NanoZoomer 2.0 RS (Hamamatsu Photonics, Shizuoka, Japan). Histological images for microscopic examination were obtained using a BZ-X710 microscope (Keyence, Osaka, Japan). The numbers and proportions of LNs and DLTs were determined after H&E staining of the whole section. Above measurement was done on minimum eight sections of each animal.

### 2.6. Immunohistochemistry (IHC) and Immunofluorescence (IF)

Neutral buffered formalin-fixed paraffin blocks were cut, and IHC and IF were performed to calculate the positive percentage of B cells (CD20), T cells (CD3), macrophages (IBA1), dendritic cells (Langerin), plasma cells (IgA and IgG), and high endothelial venules (HEVs; PNAd). Staining was performed as previously described [21]. IHC and IF staining conditions were described in Appendix A. For histoplanimetry, the total area of LNs or DLTs and the positive reaction area for the markers of B cells (CD20), T cells (CD3), macrophages (IBA1), and plasma cells (IgA and IgG) or the number of PNAd^+^ HEVs were determined based on measurements conducted on more than eight LNs or DLTs of each examined goat or pig. To calculate the positive index ratio of the immune cells, we captured digital images of immunohistochemically stained VV sections using the BZ-X710 microscope. Then, we measured LNs or DLTs area and immune cell positive area within the corresponding LNs or DLTs area using a BX-analyzer (Keyence, Osaka, Japan). Next, the average ratios of immune cells positive area/field area were calculated and compared between the two examined species. Subsequently, the positive reaction area or the number of PNAd^+^ HEVs to the total area was calculated for LNs or DLTs.

### 2.7. Scanning Electron Microscopy (SEM)

For SEM, approximately 5 × 5 mm of VV tissue containing dark blue spots as a result of hematoxylin whole-mount staining were fixed using a fixing solution containing 2.5% glutaraldehyde and 4% paraformaldehyde in 0.1 M phosphate buffer. After six washes with 0.1 M phosphate buffer, the specimens were post-fixed with 1% osmium tetroxide (OsO_4_) phosphate buffer (0.1 M) for 1 h at 4 °C. The tissue was then treated with tannic acid for 1.5 h at 4 °C. Subsequently, the specimens were dehydrated using an ascending ethanol gradient, immersed in 3-methylbutyl acetate, then dried using an HCP-2 critical point dryer (Hitachi, Tokyo, Japan). The dried specimens were sputter-coated using a Hitachi E-1030 ion sputter coater for 60 s, and then analyzed with the SEM (S-4100, Hitachi).

### 2.8. Statistical Analyses

All data are presented as mean ± standard error (SE). The Mann–Whitney *U* test was used to compare data between goats and pigs (*p* < 0.05).

## 3. Results

### 3.1. GOALTs Found in the VV of Goats and Pigs

Goat and pig VV mucosa located between the external urethral orifice and vulva was examined (Figure 1a–f). Whole-mount staining with hematoxylin was performed to visualize the LTs in the examined mucosa based on the aggregation of immune cell nuclei. Hematoxylin-positive (hematoxylin^+^), small, dark blue spots appeared on the mucosa, indicating LT, which was histologically identified after hematoxylin and eosin staining (Figure 1e,f). All examined goats and pigs showed abundant hematoxylin^+^ spots throughout the mucosal surface of the VV area, which were ring-shaped. Morphologically, we did not observe an obvious difference between goat and pig LT, but the size of the dark blue spots on the mucosal surface seemed to be larger in pigs than in goats.

### 3.2. Histological Description of GOALTs in the VV of Goats and Pigs

Histological observations were performed on the mucosa of the VV (Figure 2). Both goat and pig VV were covered with non-keratinized and stratified squamous epithelium. The basal membrane separated the epithelium and LP. However, in some cases their structure was partly or completely broken by the LTs (Figure 2a–d). LTs were predominantly distributed in the LP, but LTs bordering with lumen were also observed. Similarly to our previous results, two different types of LTs were observed in both goats (Figure 2a,c) and pigs (Figure 2b,d): VV; LN- or DLT-type. Briefly, randomly localized LNs were mostly oval-shaped and included a bright and centrally localized germinal center surrounded by densely aggregated cells (Figure 2a,b). DLTs showed an undefined shape, and these composing cells were diffusely distributed (Figure 2c,d). In the histometry of the VV mucosa, goat LNs accounted for 39.9% and pig 25.1%, goat DLT 60.1%, and pig 74.9%; pigs showed significant differences between LNs and DLTs (Figure 2e). In the VV section, we also examined the stratified squamous epithelium thickness, which was significantly thicker in goats than in pigs (Figure 2f).

### 3.3. Immune Cell Characteristics Composing GOALTs in the VV of Goats and Pigs

Various immune cells were found in goat and pig LNs of GOALT in the VV mucosa, including CD3^+^ T cells, CD20^+^ B cells, and IBA1^+^ macrophages. The CD3^+^ T cells were distributed throughout the LNs and their germinal center, as well as in the area surrounding the LNs (Figure 3a,b). The positive percentage of CD3^+^ T cells in the LN area was not significantly different between the two species (Figure 3c). The CD20^+^ B cells were densely distributed in the LN, and the positive percentage in the LN area was significantly higher in pigs than in goats (*p* < 0.05, Figure 3d–f). IBA1^+^ macrophages were similarly localized in goats and pigs, being observed in the germinal center and the area surrounding LNs (Figure 3g,h). There was no significant difference between goats and pigs regarding the positive percentage of IBA1^+^ macrophages (Figure 3i). Finally, we found that PNAd^+^ HEVs formed only around the LNs, and the numbers were not statistically different between the two species (Figure 3j–l).

Goat and pig DLT of GOALT in the VV mucosa had a similar immune cell composition to that of the LN. CD3^+^ T cells were diffusely found throughout the DLT (Figure 4a,b). In pigs, CD3^+^ T cells were found near the intraluminal space (Figure 4b). The positive percentage of CD3^+^ T cells in the DLT area was not significantly different between the two species (Figure 4c). CD20^+^ B cells were distributed thoroughly in DLT (Figure 4d,e), and the positive percentage in the DLT area was significantly higher in pigs than in goats (*p* < 0.05, Figure 4f). Diffuse IBA1^+^ macrophages were observed in the DLT (Figure 4g,h), and no significant difference was found between goats and pigs (Figure 4i). Finally, PNAd^+^ HEVs formed diffusely inside the DLT, and this number was not statistically different between the two species (Figure 4j–l).

Additionally, we checked whether the percentage of immune cells was affected by the different stages of the estrus cycle in goats (Appendix A). Although the stages of the estrus cycle in the two goats were determined using a visual assessment test, estradiol level in blood serum, vaginal cytology, vaginal temperature, vaginal pH, and VV epithelium thickness, the percentage of immune cells was not remarkably different between the estrus and diestrus stages.

### 3.4. Immunoglobulin Producing Cells and Antigen Presenting Cells in the VV of Goats and Pigs

IgA^+^ plasma cells were found within goat and pig LNs in small amounts and around the area surrounding the LN without significant differences in their quantified values (Figure 5a–c). The presence of IgA^+^ plasma cells was also observed diffusely in the DLT of both goats and pigs, without significant differences in their quantified values (Figure 5d–f). Furthermore, IgA^+^ plasma cells were aggregated to connective tissue papillae near the intraluminal space in goats (Appendix A), whereas both goats and pigs contained IgA^+^ plasma cells within the epithelium, resembling plasma cell migration from the LP of the VV to the intraluminal space (Appendix A). IgG^+^ plasma cells were distributed in the LN and DLT, similar to IgA^+^ plasma cells, and no significant difference was observed between goats and pigs (Figure 5g–l).

Additionally, we examined the localization of Langerin^+^ cells [26] and vaginal epithelial dendritic cells (VEDCs) in the LN, DLT, and VV epithelium of goats and pigs (Figure 6). No VEDCs were found in the LNs (Figure 6a,b); however, Langerin^+^ cells without dendritic cell morphology were observed in the DLT (Figure 6c,d). Abundant langerin^+^ VEDCs were found in the VV epithelium, whereas in goats, VEDCs were found in the intermediate layers of the epithelium (Figure 6e), and in pigs, VEDCs were found near the basement membrane area (Figure 6f).

### 3.5. Surface Structures of GOALTs in the VV of Goats and Pigs

Figure 7 shows the ultrastructural characteristics of GOALTs on the surface of the VV epithelium in goats and pigs. Hematoxylin^+^ spots were distributed on the VV mucosal surface (Figure 7a,b). We used SEM to observe the epithelium structure that covered the GOALTs and found partly disrupted areas where dark blue spots were formed (Figure 7c,d). Briefly, epithelial cells were flattened and detached (data not shown) around the GOALTs area. The morphology of the GOALTs surface differed from that of a normal VV surface. In the center of the modified GOALTs surface, a hole was observed beneath the epithelium, which contained abundant connective tissue fibers that created a net-like structure (Figure 7e,f) and, in goat lymphocytes, were attached to the fibers (Figure 7e); however, lymphocytes were not clearly observed in the SEM specimens of pigs (Figure 7e). Finally, all these ultrastructural features are consistent with previously reported SEM data for cow VV tissue [21].

## 4. Discussion

In this study, we compared the GOALTs of adult, female, non-pregnant goats and pigs in terms of their VV. Recently, we discovered GOALTs surrounding the VV mucosa area in a ring shape in cows [21]. This anatomical formation is titled as the genital lymphoid ring. In the present study, the genital lymphoid ring was also found in both goat and pig VVs, and their general morphological structure was consistent with previous data in cows [21]. Briefly, as summarized in Figure 8, the GOALT of VV was anatomically and histologically similar between goats and pigs, being composed of LN and DLTs, including T cells, B cells, macrophages, IgA-producing cells, and HEVs.

Species-related differences in MALT-related structures have been previously reported; palatine tonsils are absent in pigs, but those in ovine species are sequentially formed from the first day of birth [27,28,29]. In general, rodents and avian species do not develop tonsils in their oral cavities, and several tonsils are developed in specific species, such as paraepiglottic tonsils in cats, pigs, sheep, and goats. Furthermore, mice have uniform-sized follicles, whereas pigs, dogs, and ruminants have distinct and long continuous follicles in their gastrointestinal tracts [30,30]. LTs in the genital organs have also been previously introduced in pigs and goats [20,31,32]. In the VV of goats and pigs, the major type of GOALT is DLT (73%), but goats show a comparable appearance rate between LN and DLT. Cows show a higher percentage of DLT (65%) compared to LN (35%) [21], indicating that the morphology of pig GOALT differs from that of ruminants. Furthermore, both LNs and DLTs of pigs contain a significantly higher percentage of CD20^+^ B cells than those of goats. In a related report, LNs in the ileal Peyer’s patches of sheep were reported to contain approximately 77.8 ± 8.6% B cells and 3.9 ± 4.4% T cells, whereas in pigs the respective percentages were 63.1 ± 7.8% and 6.3 ± 1.3% [33,34], indicating species-related differences. Additionally, in histological sections, pig lymph nodes show inverted localization of the cortex and medulla due to their different localization from other species [34]. These morphological characteristics of lymphocyte arrangements might also affect the structural characteristics of pig MALT.

In MALTs, LNs are clearly bordered by surrounding tissues, and their germinal centers initiate activation and proliferation of lymphocytes, differentiation of plasma cells, and production of antibodies [35]. DLTs lack a clear boundary with surrounding tissue, show a formless morphology, localize between antigens, and activate an immune response [21]. Lymphocytes in the LT migrate to regional lymph nodes and support proliferation and differentiation after contact with antigens, and their progenies migrate back to the LP as effective B and T cells. Therefore, our histoplanimetrical results would reflect the species-related differences in mucosal immunity, particularly their difference between organs and antigenic exposure status, which suggests a different level of immunoreactivity in different species of animals. Additionally, GOALTs can be confused with tertiary lymphoid structures resulting from chronic infection. Ectopic lymphoid tissue can be found in inflamed non-lymphoid tissues and demonstrates the most common features of secondary lymphoid tissue [36].

The mucosa of the vagina and VV is populated by several antigen-presenting cell populations with distinct anti-inflammatory or immunological tolerance properties. Langerhans cells are a specialized DC population found in the epidermis of the skin and stratified squamous epithelium of the eye, vagina, cervix, and mouth [37,38,39]. VEDCs are major antigen-presenting cells that are important in the innate immune defense, as well as in the generation and regulation of the adaptive immune system against pathogens entering the female reproductive tract [40,41]. In the present study, we showed langerin^+^ DCs in the epithelium of the VV of both goats and pigs, which was consistent with previous studies on VEDCs in humans and several species of animals [26,38,39]. VEDCs can incorporate antigens from the lumen of the genital tract, move to the draining lymph nodes, present them to T cells, and initiate an immune reaction. This pathway of antigen-presenting cell migration from the mucosa to the nearest lymph node may represent the inductive arm of the mucosal immune system in the lower portion of the female genital tract [42,43].

Humoral immune defense is mediated by antibodies produced by highly specialized cells that synthesize and secrete abundant quantities of proteins [44]. Antibody-secreting cells are generally divided into plasmablasts and plasma cells based on their proliferation capacity [45]. In our study, we found IgA^+^ cells in the LP, epithelium, DLT, and surrounding LN. Interestingly, IgA^+^ cells seem to move from the LP to the lumen of the VV through the stratified squamous epithelium in the VV of goats and pigs. In the intestines, secretory IgA-producing cells are scattered in the intestinal LP and aggregate around intestinal crypts [46]. Therefore, similar to alimentary tracts, the VV mucosa also contains IgA-producing cells to effectively defend the mucosal immune system. Additionally, this study showed that the close-to-lumen characteristic is important for the ability of secretory IgA to form protective immunoglobulin barriers in the genital tract, which is important because when protective immunoglobulin barriers are destroyed, the genital organ is more vulnerable to pathogens.

Morphological differences in LTs between goats and pigs might be related to their estrus cycle variability or estrus stage at the time of euthanasia. However, we could not determine age-related differences in the morphological changes of GOALT in goats (Appendix A). Goats are seasonally polyestrous, short-day breeders with several estrous cycles during the fall or winter, and pigs are polyestrous and can heat more than once throughout the year. Sex hormones precisely regulate most components of the innate and adaptive immune systems in the genital tract throughout the estrous cycle. Estradiol inhibits antigen presentation by vaginal cells and those vaginal cells, which in turn influences antigen presentation, as well as B and T cell proliferation [47]. Therefore, further studies using season- and estrus cycle-matched samples are needed, and female reproductive stage- and age-related changes of GOALT in each animal species would be important for the understanding of mucosal immunity of genital organs.

## 5. Conclusions

Our observations provide a morphological basis for GOALTs in the VV of female goats and pigs. Clarification of MALTs in the external part of the genital organ contributes to finding the exact border between normal or pathological conditions, as poorly described MALT can be confused with nonspecific mononuclear cell infiltrates and tertiary lymphoid structures. This clarification is crucial for the diagnosis and treatment of female reproductive organ disorders in domestic animals.

## Figures and Tables

**Figure 1 vetsci-10-00051-f001:**
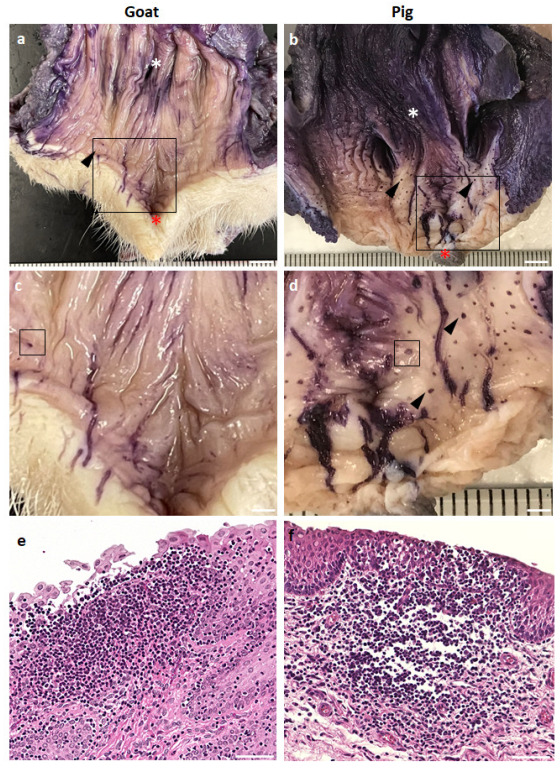
Scattered GOALTs distributed throughout the mucosa of the VV and EPGOs of goat and pig (**a**,**b**). Goat (**a**) and pig (**b**) VV and EPGOs after the whole-mount staining with hematoxylin, showing navy-blue spots. White asterisks show the external urethral orifice. Red asterisks show the clitoris. Black arrowheads indicate navy-blue spots. Scale bars = 5 mm. (**c**,**d**) Enlarged images of the VV mucosa squared areas in panels (**a**,**b**). bars = 2 mm. (**e**,**f**) Histology of navy-blue spot squared areas in panel (**c**,**d**). H&E staining. Scale bar = 50 μm.

**Figure 2 vetsci-10-00051-f002:**
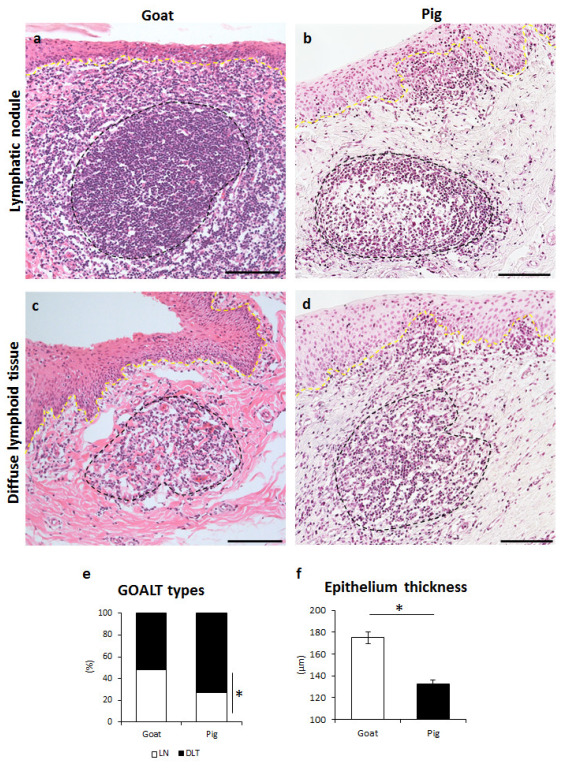
Histology of GOALTs found in the goat and pig VV. (**a**) Goat LN localizes in the LP while partially disrupting the structure of the epithelium of VV. (**b**) Pig LN localizes in the LP of VV, entirely neighbored by connective tissues. (**c**,**d**) Goat and pig DLT localizes in LP of VV not disrupting epithelium. Diffusely distributed lymphocytes are shown between DLT and epithelium. Black dotted line shows the approximate boundary of GOALTs. Yellow dotted line indicates basement membrane. H&E staining. Scale bars = 100 μm. (**e**) Appearance of the percentage of LNs and DLTs in the examined GOALTs of the VV. *n* ≥ 5. (**f**) Epithelium thickness of VV. Values = mean ± SE. *n* ≥ 5. Significant difference between the goat and pig is indicated by * *p* < 0.05; Mann–Whitney *U* test.

**Figure 3 vetsci-10-00051-f003:**
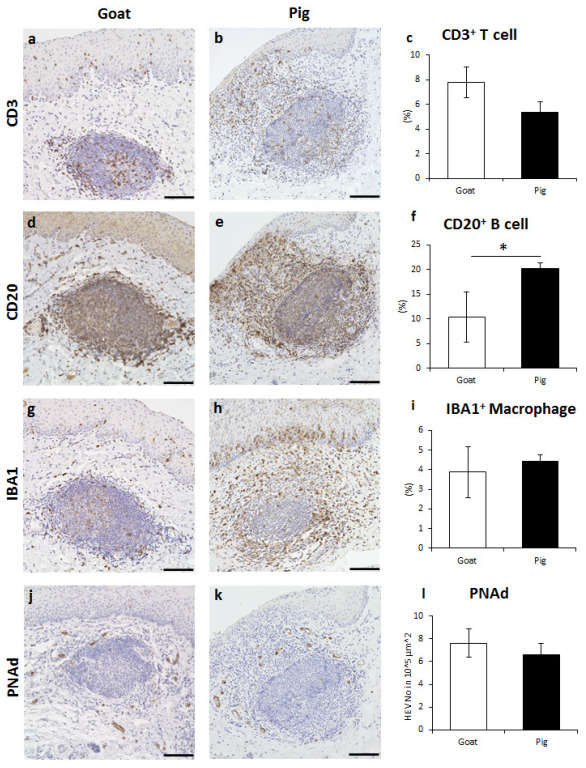
Immune cell characteristics composing LN in goat and pig VV. (**a**,**b**) CD3 for T cells of goat and pig. (**c**) Positive percentage of CD3^+^ T cells localizing in the LN. Values = mean ± SE. *n* ≥ 4. (**d**,**e**) CD20 for B cells of goat and pig. (**f**) Positive percentage of CD20^+^ B cells localizing in the LN. Values = mean ± SE. *n* ≥ 4. Significant difference between the goats and pigs is indicated by *, *p* < 0.05; Mann–Whitney *U* test. (**g**,**h**) IBA1 for macrophages of goat and pig. (**i**) Positive percentage of IBA1^+^ macrophages occupying the LN. Values = mean ± SE. *n* ≥ 4. (**j**,**k**) PNAd for HEVs of goat and pig. (**l**) Number of PNAd^+^ HEVs in 10^5^ μm^2^ in LN. Values = mean ± SE. *n* ≥ 4. Scale bars = 100 μm.

**Figure 4 vetsci-10-00051-f004:**
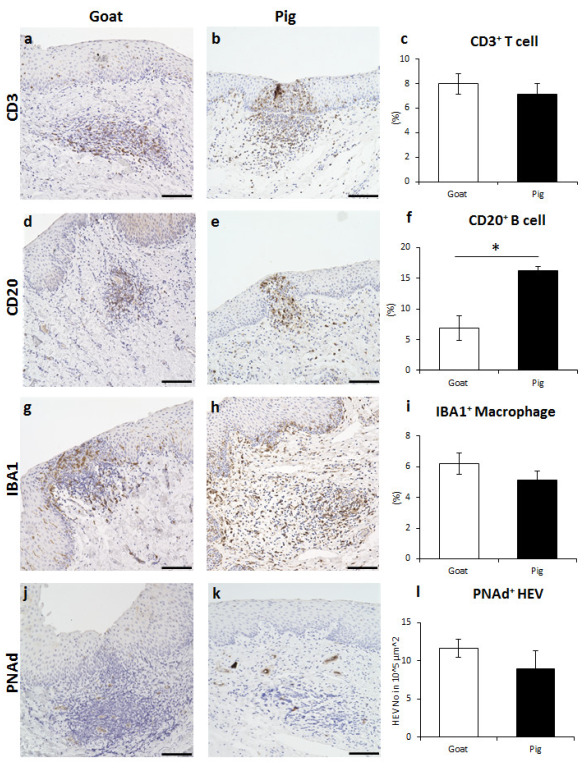
Immune cell characteristics composing DLT in goat and pig VV. (**a**,**b**) CD3 for T cells of goat and pig. (**c**) Positive percentage of CD3^+^ T cells localizing in the LN. Values = mean ± SE. *n* ≥ 4. (**d**,**e**) CD20 for B cells of goat and pig. (**f**) Positive percentage of CD20^+^ B cells localizing in the LN. Values = mean ± SE. *n* ≥ 4. Significant difference between the goats and pigs was indicated by *, *p* < 0.05; Mann–Whitney *U* test. (**g**,**h**) IBA1 for macrophages of goat and pig. (**i**) Positive percentage of IBA1^+^ macrophages occupying the LN. Values = mean ± SE. *n* ≥ 4. (**j**,**k**) PNAd for HEVs of goat and pig. (**l**) Number of PNAd^+^ HEVs in 10^5^ μm^2^ in LN. Values = mean ± SE. *n* ≥ 4. Scale bars = 100 μm.

**Figure 5 vetsci-10-00051-f005:**
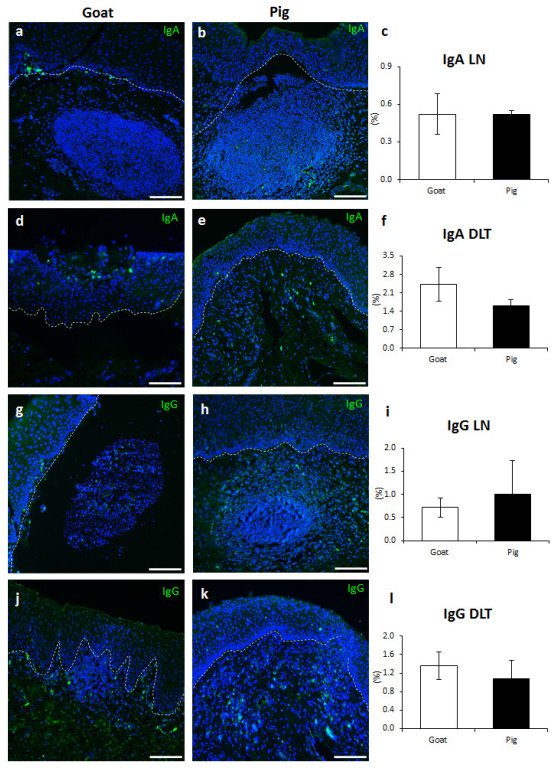
IgA^+^, IgG^+^ plasma cell characteristics in GOALT of goat and pig. (**a**,**b**) IgA^+^ plasma cells in the LN of goat and pig. (**c**) Positive percentage of IgA^+^ plasma cells localized the LN. Values = mean ± SE. *n* ≥ 4. (**d**,**e**) IgA^+^ plasma cells in the DLT of goat and pig. (**f**) Positive percentage of IgA^+^ plasma cells localized the DLT. Values = mean ± SE. *n* ≥ 4. (**g**,**h**) IgG^+^ plasma cells in the LN of goat and pig. (**i**) Positive percentage of IgG^+^ plasma cells localized the LN. Values = mean ± SE. *n* ≥ 4. (**j**,**k**) IgG^+^ plasma cells in the DLT of goat and pig. (**l**) Positive percentage of IgG^+^ plasma cells localized the DLT. Values = mean ± SE. *n* ≥ 4. Dotted lines indicated the approximate boundary between LP and epithelium. Scale bars = 100 μm.

**Figure 6 vetsci-10-00051-f006:**
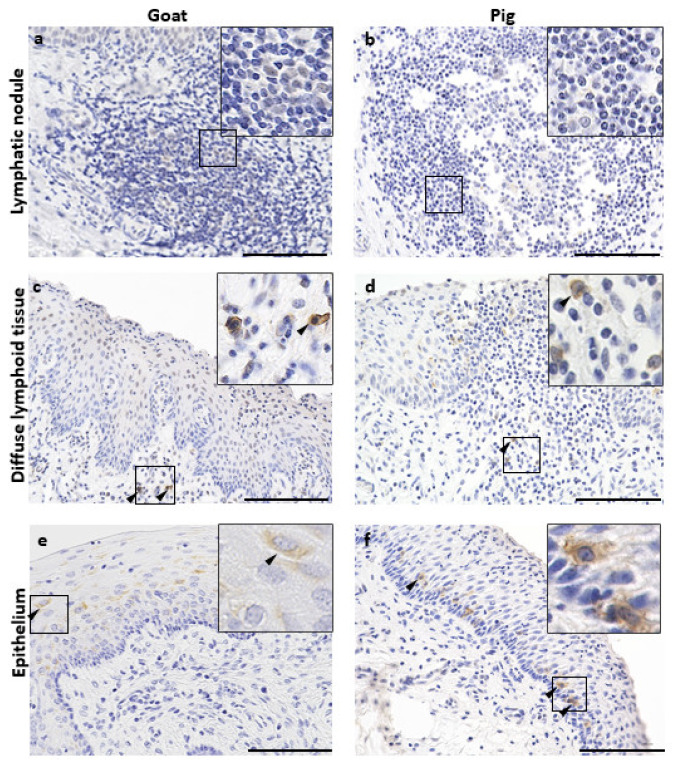
Dendritic cells characteristic in GOALT and epithelium of VV of goat and pig. (**a**,**b**) Langerin^+^ cells in the LN of goat and pig. (**c**,**d**) Langerin^+^ cells in the DLT of goat and pig. (**e**,**f**) Langerin^+^ cells in the of goat and pig. Black arrowheads show positive reaction for IHC. Scale bars = 100 μm.

**Figure 7 vetsci-10-00051-f007:**
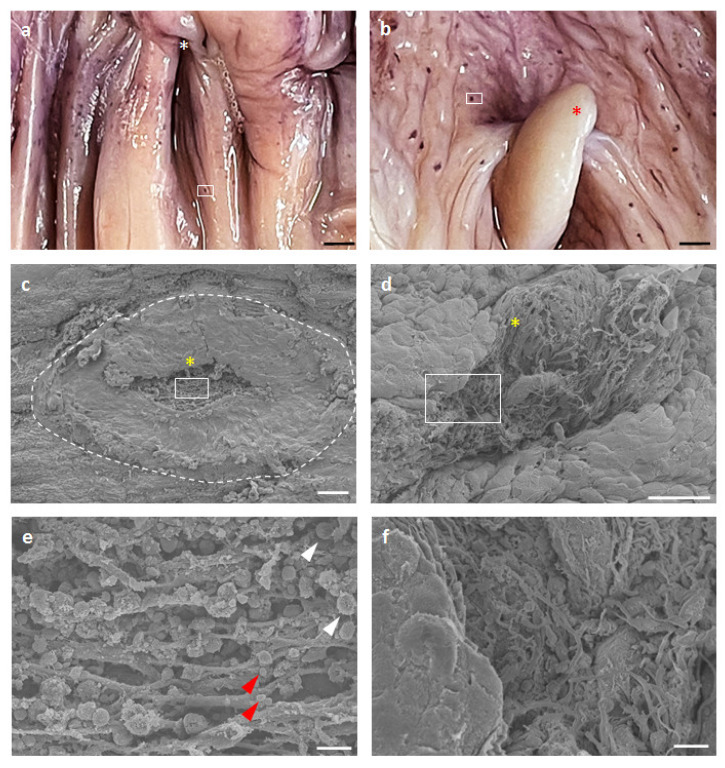
Ultrastructural characteristics of epithelium covering GOALT of the goat and pig. (**a**,**b**) Goat (**a**) and pig (**b**) VV mucosa stained in whole-mount with hematoxylin. GOALTs were observed as hematoxylin-positive blue spots. Scale bars = 2 mm. (**c**,**d**) Goat (**c**) and pig (**d**) VV mucosa were examined using SEM. The same areas as in panels (**a**,**b**) were examined. Dotted lines indicate GOALTs appeared on the surface of the epithelium. Yellow asterisks represent GOALTs opening into the VV lumen. Scale bars = 100, 50 μm. (**e**,**f**) Magnified areas squared in panels (**c**,**d**). White arrowheads indicate infiltration of lymphocytes from the GOALT to the lumen of the VV. Red arrowheads indicate erythrocyte. Red asterisk shows clitoris. Scale bars = 10 μm.

**Figure 8 vetsci-10-00051-f008:**
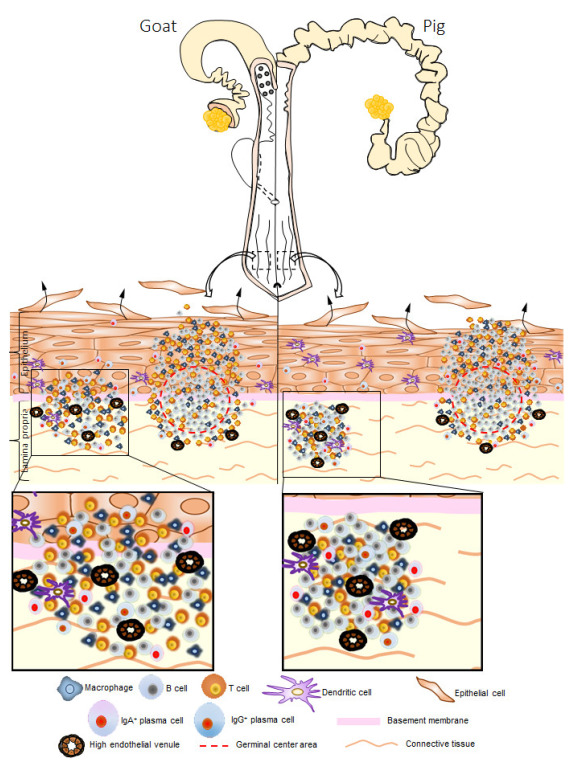
Graphical representation of GOALTs in the VV and reproductive organs of goat and pig. VV is one of the compartments of the EPGO containing GOALTs. GOALTs consist of LNs and DLTs which localize in different parts of LP of the VV. LN and DLT comprised of B cells, T cells, macrophages, and APCs. B cells are predominantly found in both LN and DLT of pigs. Germinal center mainly found in the center of the LN. HEVs observed among LT and neighboring connective tissues in the LP. Basically, the epithelium structure covered GOALTs is partly or thoroughly disrupted by the invasion of immune cells composing LNs and DLTs. LNs and DLTs are arranged as ring in the VV mucosa and titled as “genital lymphoid ring”.

## Data Availability

The datasets generated and/or analyzed during the current study are available from the corresponding author upon reasonable request.

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
