# Peer review of "Morphological Characteristics of Genital Organ-Associated Lymphoid Tissue in the Vaginal Vestibule of Goats and Pigs"

_vetsci, 2023, doi:10.3390/vetsci10010051_

Round 1

Reviewer 1 Report

This study shows unique results by analyzing the lymphoid tissue of the pubic region of goats and pigs. It will be commended as the fundamental research to achieve animal characteristics and veterinary management of domestic animals. I would like to present a few questions and modifications as follows. Please reflect the author's response in the text if necessary

1.       L72-74: Do you think the character of a long urethra and suburethral diverticulum in pigs is related to the development of GOALT?

2.       Fig. 3b, 3e, 3h: Does the infiltration of lymphocytes in the epithelium imply a state of increased immune activity? Were there any individual differences in these results among the eight pigs?

3.       Fig. 3c, 3f, 3i, and Fig. 4c, 4f, 4i: Please specify what cells were the parent population for calculating these percentages. Please specify the calculation process in the material and method for the percentages adding what types of cells were counted.

4.       It is hard to understand the difference in contents between Fig. 3 and Fig. 4. Please use a different title for each figure.

5.       L304 “round-shaped morphology”: The cell indicated by the arrow in Fig. 6c is not round.

6.       Fig. 7: Are there any differences in what authors wanted to show between Fig. 7a-b and Fig. 1a-d? If so, please describe it in the text.

7.       Fig. 7: Did you forget the white arrowheads in Fig. 7d? Or is it because you could not observe lymphocytes in the GOALT opening? If the latter, please describe it in the text.

8.       L357-358: You described that pigs have a long VV in the Introduction. Was there a preferred site for the lymphoid ring? (near the external urethral opening, near the labia, etc.). If you could identify the specific location of the ring-shaped hematoxylin spots, please add them to the description.

9.       L385-387: Please express this conclusion in more detail: does it mean that the immunoreactivity in GOATLTs in VV is higher in pigs than in goats?

10.    Supplemental figure 1: Why is there a period of increasing superficial cells during the diestrus period?

11.    Do you believe that the lymphocytes in DLT and LN are in different active states? Do you have evidence that DLT and LN types migrate to each other?

Minor comment

1.       L118: Remove “by the”

2.       L140: Add the staining methods for observation of vaginal smears.

3.       L165-167: What is the difference in usage between using NanoZoomer 2.0 RS and BZ-X710 microscope?

4.       L176: Please describe the means of measurement for “histoplanimetry” (e.g., software names)

5.       Supplementary Tables 1 and 2: Indicate the product number or clone number for used antibodies for readers’ accessibility

6.       L172 NBF: The words should be shown by full spelling at the first appearance

Author Response

Comments from Reviewer 1

Comment 1. L72-74: Do you think the character of a long urethra and suburethral diverticulum in pigs is related to the development of GOALT?

Response: Thank you for your interesting suggestion. Based on several evidence, we consider that there is no relation among long urethra, suburethral diverticulum, and the development of genital organ-associated lymphoid tissues (GOALTs) in pigs. Although the suburethral diverticulum is observed in cows and pigs, GOALTs were also found in other animal species which do not develop the suburethral diverticulum. Furthermore, we examined GOALTs of vaginal vestibules (VVs) in several species including cows and pigs as well as goats, dogs, and camels (data is not published), and all animals developed GOALTs regardless of the length of the urethra and existence or absence of suburethral diverticulum.

Comment 2. Fig. 3b, 3e, 3h: Does the infiltration of lymphocytes in the epithelium imply a state of increased immune activity? Were there any individual differences in these results among the eight pigs?

Response: As reviewer mentioned, we also considered that lymphocyte infiltrations in the epithelium reflect a state of increased immune activity. On the other hand, all examined animals in the present study were clinically diagnosed as healthy. Therefore, infiltrations of immune cells in the epithelium would reflect the normal immune function of GOALTs in VVs of examined animals. Furthermore, we could not find large individual differences in examined pigs.

Comment 3. Fig. 3c, 3f, 3i, and Fig. 4c, 4f, 4i: Please specify what cells were the parent population for calculating these percentages. Please specify the calculation process in the material and method for the percentages adding what types of cells were counted.

Response: Thank you for the valuable suggestions. We have specified cell types in Fig. 3c, 3f, 3i, and Fig. 4c, 4f, and 4i in the revised manuscript. Therefore, we have added the calculation methods and cell types which were counted as percentages (P4, L189-194).

Comment 4. It is hard to understand the difference in contents between Fig. 3 and Fig. 4. Please use a different title for each figure.

Response: We apologize for the mistake about wrongly putting the title for Fig.4. The correct title is “Immune cell characteristics composing DLT in goat and pig VV” which is included in the revised manuscript (P14, L304).

Comment 5. L304 “round-shaped morphology”: The cell indicated by the arrow in Fig. 6c is not round.

Response: We agree with the reviewer’s comment and have edited the sentence in the revised manuscript as “No VEDCs were found in the LNs (Figure 6a, b); however, Langerin+ cells without dendritic cell morphology were found in the DLT” (P15, L325-327).

Comment 6. Fig. 7: Are there any differences in what authors wanted to show between Fig. 7a-b and Fig. 1a-d? If so, please describe it in the text.

Response: Thank you for the suggestion. Fig. 1a-d and 7a-b are both same whole-mount staining with hematoxylin. Figures showed different individual VV mucosal surfaces; however, there was no obvious differences. In Fig.1, we proposed to show hematoxylin-positive spots on the mucosal surface are which indicating lymphoid tissues. In Fig. 7, we tried to make readers easy to understand the ultrastructure of lymphoid tissues of hematoxylin-positive spots on the VV surface.

Comment 7. Fig. 7: Did you forget the white arrowheads in Fig. 7d? Or is it because you could not observe lymphocytes in the GOALT opening? If the latter, please describe it in the text.

Response: Thank you for the indications. There were no clear lymphocytes within the GOALT opening of the pig in SEM specimens. We have added reviewers’ opinions in the revised manuscript (P18, L356-358).

Comment 8. L357-358: You described that pigs have a long VV in the Introduction. Was there a preferred site for the lymphoid ring? (near the external urethral opening, near the labia, etc.). If you could identify the specific location of the ring-shaped hematoxylin spots, please add them to the description.

Response: Thank you for your valuable comments. Although we had investigated the specific site to form the lymphoid ring in goat and pig VV, there was no specific localization. In several animals, GOALTs were found around the external urethral opening, while in others GOALTs were observed near to clitoris area. In addition, our previous study investigated the localization of GOALTs in cow VVs, which showed no specific localization of GOALTs (Chuluunbaatar, T. et. al., Genital Organ-Associated Lymphoid Tissues Arranged in a Ring in the Mucosa of Cow Vaginal Vestibules. Research in Veterinary Science, 2022). 

Comment 9. L385-387: Please express this conclusion in more detail: does it mean that the immunoreactivity in GOATLTs in VV is higher in pigs than in goats?

Response: Thank you for your comments. We have described more in detail in the revised manuscript as possible. CD20+ B cells from pigs might reflect the enhanced immunoreactivity compared with goats; however, we could not exactly express that GOALTs in VV of pig has higher immunoreactivity compared with goats (P21, L417-418).

Comment 10. Supplemental figure 1: Why is there a period of increasing superficial cells during the diestrus period?

Response: We apologize to the reviewer for any misunderstanding. Supplemental figure 1-I is the vaginal smear result of the goat throughout 23 days. Our hypothesis was 23rd day (indicated with a red arrow) is the beginning of the diestrus cycle which is after a sudden decrease of superficial cells and an increase of intermediate cells. 

Comment 11. Do you believe that the lymphocytes in DLT and LN are in different active states? Do you have evidence that DLT and LN types migrate to each other?

Response: Thank you for your indication. Firstly, we are considering that DLT and LN would be in different active states. Because several LNs possess a germinal center, indicating the activation of adaptive (antigen-specific) immunity. The site where antigen-presenting cells actively interact with naive B-cells to initiate an antigen-specific immune response. On the other hand, DLTs did not form a follicle-like structure with the germinal center, and lymphocytes and plasma cells localized diffusely in the lamina propria. However, we consider that they have a common function in the point of an immunological barrier against the invasion of microorganisms.

We apologize to the reviewer for any misunderstanding. We tried to mean that “Migrating” is immune cell migration from lymphoid tissue to the epithelium; therefore, we do not have exact evidence that DLT- and LN- types migrate to each other.

Minor comment

Comment 1. L118: Remove “by the”

Response: We have removed mentioned part from the revised manuscript.

Comment 2. L140: Add the staining methods for observation of vaginal smears.

Response: We have added a staining method for observation of vaginal smear (P4, L150-153).

Comment 3. L165-167: What is the difference in usage between using NanoZoomer 2.0 RS and BZ-X710 microscope?

Response: NanoZoomer 2.0 RS is a digital slide scanner that converts glass slides into high-resolution digital data by high-speed scanning. NanoZoomer scanned sections are convenient to use anytime and from anywhere for a long time regardless of having a microscope. BZ-X710 microscope is a compact-sized microscope that supports fluorescence, brightfield, and phase contrast imaging. We used the BZ-X710 microscope for highly qualified pictures and immunofluorescence observation.

Comment 4. L176: Please describe the means of measurement for “histoplanimetry” (e.g., software names)

Response: We have described histoplanimetrical measurements (P4, L189-194).

Comment 5. Supplementary Tables 1 and 2: Indicate the product number or clone number for used antibodies for readers’ accessibility.

Response: According to the reviewer’s suggestion, we have added the product number for used antibodies in Supplementary Tables 1 and 2.

Comment 6. L172 NBF: The words should be shown by full spelling at the first appearance.

Response: We have edited NBF to neutral buffered formalin (P4, L180).

We appreciate your careful reading again.

Reviewer 2 Report

     The data demonstrated in this study appears to include the considerable data for the lymphoid tissue in the vaginal vestibule of goats and pigs, that should provide the grate benefits to the anatomists, immunologists and clinical veterinarians. Some points should be clarified more and revised.

1) This study argued that the number of CD20-positive cell (B cell) in the lymphoid tissue was significantly different between goats and pigs. On the other hand, even in pigs, CD20-positive cell occupied only 20% in this tissue. So what types of cells occupies remaining large part of this lymphoid node?

2) The authors should clarify their opinion whether anti-CD20 antibody used in this study detect almost B cells or only a subset of B cells. If this antibody detects only some subsets, there is a possibility that the demonstrated data indicates the different composition of subset, not the number of B cell, between goats and pigs.

3) The illustration of figure 8 is not much to the data demonstrated in this study. T cell composition between goats and pigs is not significantly different, but the illustration of goat lymphoid node contains the significant number of T cell comparing to that of pig lymphoid node. To avoid readers misunderstanding, this illustration should be revised.

4) Please check the manuscript carefully, again.

(1)In line 118, “by the” should be deleted.

(2)In line 126, is double-space inserted between words “and” and “tissue”?

(3)In fig2A, there is no figure lettering “a” on the photograph.

(4) In the legend of figure 7, the red asterisk should be explained. “Red arrowhead” should be “Red arrowheads”.

(5) Please check the reference style again, especially ref. No. 36 and 46.

Author Response

Comments from Reviewer 2

Comment 1. This study argued that the number of CD20-positive cell (B cell) in the lymphoid tissue was significantly different between goats and pigs. On the other hand, even in pigs, CD20-positive cell occupied only 20% in this tissue. So, what types of cells occupies remaining large part of this lymphoid node?

Response: We appreciate your indications. The significant difference in CD20-positive B cells between goats and pigs would be related to the species-specific difference. For the meaning of “20 %”, we used a BX-analyzer (Keyence, Osaka, Japan) which can quantify the positive reaction of immunohistochemistry.  Furthermore, several immune-related cells, including antigen-presenting cells, were not checked by the immunohistochemistry staining, these cells would be a component of lymphatic nodules (LNs) and diffuse lymphoid tissues (DLTs). These factors finally affect the results of histoplanimetry.

Comment 2. The authors should clarify their opinion whether used in this study detect almost B cells or only a subset of B cells. If this antibody detects only some subsets, there is a possibility that the demonstrated data indicates the different composition of subset, not the number of B cell, between goats and pigs.

Response: Thank you for raising this point. Actually, CD20 is used as a pan-B cell marker although a paper indicated the expression of a minor subset of T cell (Hultin L.E. et. al., CD20 (pan-B cell) antigen is expressed at a low level on a subpopulation of human T lymphocytes. Cytometry. 1993). We have confirmed this antibody usage and staining condition on dogs and cats (Škor, O., al., Are B-symptoms more reliable prognostic indicators than substage in canine nodal diffuse large B-cell lymphoma. Veterinary Comparative Oncology. 2021; Ichii O., et al., Ureteral morphology and pathology during urolithiasis in cats, Research in Veterinary Science, 2022)

Comment 3. The illustration of figure 8 is not much to the data demonstrated in this study. T cell composition between goats and pigs is not significantly different, but the illustration of goat lymphoid node contains the significant number of T cell comparing to that of pig lymphoid node. To avoid readers misunderstanding, this illustration should be revised.

Response: Thank you for the suggestion. We have included the reviewer’s opinion in the revised manuscript Fig. 8.

Comment 4: Please check the manuscript carefully, again.

  • In line 118, “by the” should be deleted.
  • In line 126, is double-space inserted between words “and” and “tissue”?
  • In fig2A, there is no figure lettering “a” on the photograph.
  • In the legend of figure 7, the red asterisk should be explained. “Red arrowhead” should be “Red arrowheads”.
  • Please check the reference style again, especially ref. No. 36 and 46.

Response: Thank you for the careful checking. All comments are reflected in the revised manuscript.

Reviewer 3 Report

Dear Authors,

I have read your manuscript with interest. In general, it is quite well written and easy to understand. Here and there, I have found a type or think that the sentence should be rephrased as it is not always clear. In addition, I have some questions and comments regarding the content. Please find my limited number of comments in the attached, scanned manuscript onto which I have written my remarks.

Author Response

Comments from Reviewer 3

Comment 1: P1, L40: What you mean? “Intercellular space”

Response: Vaginal vestibule epithelium covering genital organ-associated lymphoid tissue structure was fully or partially broken in result of disruption of tight junction and comparatively large space was observed between epithelial cells and we entitled it as “intercellular space”, which we showed in our previous article (Chuluunbaatar, T. et. al., Genital Organ-Associated Lymphoid Tissues Arranged in a Ring in the Mucosa of Cow Vaginal Vestibules, Research in Veterinary Science, 2022).

Comment 2: P2, L53-54: Is the M-cells a mobile cell type?

Response: Thank you for your accurate review. M-cells are not mobile cell types. This was our mistake. We have excluded M-cells from the cells infiltrating the epithelium (P2, L56).

Comment 3: P2, L57: Lymphoid tissues of Waldeyer’s ring belong either to the MALT, GALT, or LALT (of a review article by Casteleyn et al.)

Response: We fully agree with the reviewer’s suggestion that the lymphoid tissues of Waldeyer’s ring belong to the MALT. Generally, Waldeyer’s ring consists of the pharyngeal tonsil, tubal tonsil, tonsil of the soft palate, palatine tonsil, and lingual tonsil (Liebler-Tenorio E., et. al., MALT structure and function in farm animals. Veterinary Research, 2006). Therefore, Waldeyer’s ring partially belongs either to the GALT or LALT.

Comment 4: P10, L282: What is the difference with Fig. 3?

Response: We apologize for the inattention. We have corrected the title for Fig. 4 as “Immune cell characteristics composing DLT in goat and pig VV” (P14, L305).

Comment 5: P16, L373-374: Where does this huge difference come from?

Response: Thank you for your accurate review. It was our typing mistake. We have corrected the percentage number in the manuscript. B cell percentage in ileal Peyer patches of a pig is 63.1±7.8%.

Comment 6: P16, L395: How is this like since M cells vary in shape in various species?

Response: Thank you for pointing this out. In the gastrointestinal tract, M-cells play an important role in the transport of antigens from the lumen of the small intestine to mucosal lymphoid tissues, where processing and initiation of immune responses occur. Also, in the skin, the local Langerhans cells take up and process microbial antigens as functional antigen-presenting cells. We hypothesize that langerin positive dendritic cells in the vaginal vestibule can act as M-cells or Langerhans cells. To not misleading readers, we have deleted the phrase “M cell-like” (P22, L431).

In addition to the above comments, all spelling and grammatical errors pointed out by the reviewer have been corrected.